# Study on the Urban Fringe Based on the Expansion–Shrinking Dynamic Pattern

**Changqing Sui and Wei Lu \***

School of Architecture and Art, Dalian University of Technology, Dalian City 116024, China; scq@mail.dlut.edu.cn (C.S.)

**\*** Correspondence: luweieds@dlut.edu.cn; Tel: +86-0411-8470-8530

**Abstract:** The urban fringe, as a part of an urban spatial form, plays a considerably major role in urban expansion and shrinking. After decades of rapid development, Chinese cities have advanced from a simple expansion stage to an expansion–shrinking-coexistence stage. In urban shrinking and expansion, the urban fringe shows different characteristics and requirements for specific aspects such as urban planning, land use, urban landscape, ecological protection, and architectural form, thereby forming expanding and shrinking urban fringes. A comprehensive study of expanding and shrinking urban fringes and their patterns is theoretically significant for urban planning, land use, planning management, and ecological civilisation construction.

**Keywords:** urban fringe; urban growth; urban shrinking

## 1. Introduction

With its reform and opening up in the late 1970s, the industrialisation and urbanisation of China have entered a rapid development stage. In the early stage of reform and opening up, China's urbanisation rate was only 17.92% (1978), which increased to 60% in 2019 (*2020 Social Blue Book* issued by the Chinese Academy of Social Sciences). According to international standards, this phenomenon indicates that China has realised urbanisation. With continuous and high-speed urbanisation, especially after 1990, China based its urban planning on growth logic[1]. However, due to China's large territory and substantial differences on the basis and level of urban development, urbanisation shows a complex phenomenon. Some cities have continued to grow, showing typical urban expansion, and other cities, such as some cities in the northeast, have experienced urban shrinking. A paradox of urban shrinking is that urban population is shrinking, but urban construction is expanding [2]. Chinese cities have transformed from the simple expansion stage to the expansion–shrinkage stage.

By contrast, an urban fringe is a part of a city, which is inevitably involved in urban expansion and shrinking. Land use, spatial structure, ecological landscape, planning and management, and industrial development in the urban fringe[3] have always been the concerns of related scholars. Chinese scholars began studying the urban fringe in the late 1980s. In 1989, Gu Chaolin studied the urban fringe for the first time and reported that 'one of the most obvious characteristics of urbanisation is advancement near the urban area and wide-area expansion. When it is reflected in urban regional structures, the most sensitive and rapidly changing areas of urbanisation are in the urban fringe' [4].

The urban fringe became a sought-after research topic in 2005, and the attention given to this field began to decrease in 2015. The focus of studies on the urban fringe based on the urban expansion logic is in line with the urban expansion rhythm. During 2014–2015, when the attention given to this field decreased significantly, the urban shrinking phenomenon received the attention of Chinese scholars. This phenomenon suggests that the studies on urban shrinking should be based on the urban shrinking perspective to

render the investigation structure of the urban fringe more balanced and complete than China's urbanisation and to bring it in line with the regularity and demand of China's urbanisation [4].

## 2. Urban Fringe

### 2.1. History of the Urban Fringe

Before the German geographer H. Louis first proposed the urban fringe concept in 1936, the concentration of cities and dispersion of their periphery presented their own advantages and disadvantages in production and life. Although the urban fringe concept did not exist before 1936, cities did have fringes. Mumford reported the following about suburbs: 'But the fact is that suburbs appeared considerably early similar to cities. Woolley found the evidence to prove that the City of Ur has suburbs outside the developed area'[5]. Mumford also reported that Alberti's architectural theory presented the aesthetic and psychological advantages of suburbs [5]. In China, the princes of Wu and Yue constructed palaces, farewell halls, and banquet places in the suburbs in Spring and Autumn. These constructed sites were the earliest places for scenic recreation activities in history. In the eastern suburbs of the Chang'an City in the Tang Dynasty, nobles built many private villa gardens. In the southern suburbs, literati and officialdom gardens are widely distributed[6].

In 18th-century Europe, industrialised cities emerged. Compared with the industrialised urban centres, irrespective of whether they are early Coketown [5] or later industrial centres, their fringe areas are beautiful suburbs. The aforementioned general analysis indicates that the urban fringe presents highly valuable resources such as land, space, and landscape for the wealthy and powerful class, and for the low class, the urban fringe area is only a garbage dump that pollutes production and low-cost shelters away from various social resources available in city centres.

Japanese scholars Kato and Haruka devoted themselves to the study of urban fringe areas[7,8]. They studied the walkability of urban fringe areas in 2020, and analysed the relationship between the location of urban facilities and population decline in the Osaka metropolitan fringe area in 2021. Perrin, Coline, Clement, and Camille[9] reviewed the farmland protection and land-use planning in urban fringe areas and explored the suburban farmland protection and land use in the Mediterranean area.

### 2.2. Concept, Essence, and General Characteristics of the Urban Fringe

No absolute unified understanding of the urban fringe is available; however, this does not substantially affect urban fringe studies. The urban fringe concept was first proposed by the German geographer H. Louis.

Domestic scholars have paid attention to the importance of urban fringes in many aspects and have studied these fringes from the following perspectives: (1) Spatial morphology: the continuity of a subject and external heterogeneous interface produces spatial differentiation, thereby forming a clear boundary or cross mixing. Considering the boundary as a dividing line, the space area within a certain range inside and outside the dividing line can be regarded as the fringe area. The parts with clear boundaries can be divided into outer and inner fringe areas. The crisscrossed fringe area can be called a mixed fringe area. (2) Administrative management: the management boundary is divided into urban and suburban areas for the space and region. The urban fringe is a contact zone of urban and suburban areas. (3) Socioeconomic landscape characteristics: the urban fringe is the junction between the urban and rural economies, societies, and landscapes or the intersection of urban and rural environmental landscapes.

*2.3. Characteristics of the Urban Fringe in China*

Cui Gonghao and Wu Jin found that China's urban fringe did not follow the suburbanisation of foreign cities but had its own characteristics. Their study divided the development stage characteristics of China's urban fringe into agricultural, semi-industrial, and industrial stages [10]. In the late stage of urbanisation, the development of spatial structure of the urban fringe shows the characteristics of comprehensive type, and many development zones, new towns, and bonded areas show the common development of industries, residence, and consumption. In this manner, the development of China's urban fringe entered the fourth stage, that is, the comprehensive development stage. In the agricultural stage, the urban fringe (outer fringe) is rural, indicating backwardness. In the semi-industrial and industrial stages, the urban fringe presents the embodiment of dirty, messy, and poor areas. In the comprehensive stage, due to the implantation of residential function and demands of commercial environment, urban fringe integrity is gradually aligned with the urban standard. Comprehensive development is often driven by top-down urban growth forces[11], in which urban planning management plays an important role. However, according to the bottom-up mechanism of basic-level organisations such as townships, villages, and individuals in the urban fringe, spontaneous urbanisation shows a different picture. This type of situation is highly complex. In addition to an increase in township enterprises with positive significance, numerous illegal buildings and many low-cost residents involved in low-end industries appeared. The poor living environment constitutes the overall environment of the urban fringe, and the fringes of large cities are even worse. In 1993, Gu Chaolin and Chen Tian et al. [12] studied the population and social characteristics of the urban fringe. Moreover, the development status of large cities such as Beijing subsequently confirmed the study conclusions of the concentration of floating population and complex social relations.

The spatial expansion of China's urban fringe is closely related to industrialisation. The strategy of 'shifting from labour-intensive industries to service economy' is implemented in cities, and the urban fringe has become the most important bearing area of all types of industries. The market logic is the basic logic of spatial expansion and land use in the urban fringe, and the core of this market logic is efficiency. As a result of pursuing profit maximisation, low-cost land is prioritised for land use, and the high-cost lands become an island far away from capital, which is the fundamental reason for the emergence of numerous urban villages (interest game) and for the leaping and networking of space expansion.

## 3. Analysis of Urban Shrinking and Growth and Factors Influencing Their Growth and Shrinking

*3.1. Growth and Shrinking*

The development of cities shows the coexistence of expansion and shrinking. The main factors affecting urban expansion and shrinking are population, economic activities, and spatial environmental quality. When the economy prospers and population increases, the urban space expands, and when the economy is depressed and population decreases, the urban space shrinks. Good spatial environmental quality attracts people, and bad spatial environmental quality leads to population loss or local migration, resulting in the overall contraction of cities or local fluctuations. Excessive urban growth leads to several urban and ecological problems. In the 1960s, the developed countries in Europe and America experienced urban economic growth weakness, leading to an increase in unemployment rates, deterioration of the living environment, and massive population loss, which was termed as 'urban shrinking' by scholars [13]. Since the 1970s, numerous studies on growth doctrine have been conducted. Afterwards, theories or issues such as 'limits of growth', 'sustainable development', 'spreading cities', 'smart growth', 'smart shrinking', and 'tightening cities' were proposed. In the urbanisation process in China, both the market and government are constantly enhancing growth awareness. Land use

and spatial planning and management based on sustainable growth reflect the thinking and behaviour patterns of growth doctrine, and the reflection on growth in China has just begun.

Between 1990 and 2010, 20% of European cities experienced shrinkage. Until now, 883 cities are suffering from or facing the problem of shrinkage [14]. The shrinking city international research network (SCIRN), composed of many researchers, aims to learn about different urban shrinking methods and the effects of these policies and strategies on urban regeneration [15]. Wiechmann and Thorsten [16] analysed the transformation patterns and local strategies after urban shrinkage in Germany and the United States. Berglund and Lisa [17] discussed Detroit's response to shrinkage based on economic adjustment and talent cultivation. In addition, some researchers argued that many cities in Europe and the United States must cope with the challenges posed by long-term demographic and economic changes [18].

The ultimate cause of urban growth is the continuous growth of people's requirements. The consumer culture resulting from industrialisation has spread globally, changing people's perception towards consumption. Expanding consumption stimulates production expansion, but simultaneously, it expands the ecological environment problems. In the era of agricultural civilisation, economic growth is limited, and in the era of industrial civilisation, the development of science and technology repeatedly breaks through the limits set by nature, rendering people to have the illusion of unlimited development.

The urban fringe development based on the growth logic presents several characteristics and demands: (1) the principle of maximising local economic benefits and overall extensive (structural income tendency) development; (2) risk, optimism, and opportunism; (3) the imbalance of political and economic game forces; (4) domination by local government performance; and (5) single thinking (GDP doctrine).

Cities cannot grow indefinitely, and growing cities may face shrinking. Resource depletion, industrial transfer, recession, and population decline lead to a decrease in the urban operation efficiency and increase in costs, and thus the city shrinks. Generally, urban population loss is used as the basis for urban shrinking; however, the SCIRN advocates that urban shrinking should be defined from multi-dimensional perspectives, such as economic recession [15]. In urban development and change identification, in addition to the traditional statistical data, studies have used night-light data [19]. Domestic scholar Zhang Jingxiang summarised three patterns of urban shrinking [20]: trend-, overdraft-, and adjustment-type shrinking. Trend-type shrinking is long-term shrinking, and overdraft- and adjustment-type shrinking are short-term shrinking. Since the 1990s, the old industrial bases in Northeast China began to slowly develop or shrink, and after 2000, numerous cities in Northeast China shrunk. Zhang Mingdou reported that 'according to the growth rate of the total population, 13 shrinking cities have been identified, accounting for 31.71% cities in Northeast China. From 2000 to 2010, the urban population loss in Northeast China has been severe, and large-scale urban shrinking has already occurred' [21]. His research divided urban shrinking into global, central, and fringe shrinking [21]. In 5 years (2014–2018), the population of Changchun and Harbin, the provincial capitals, decreased, and that of Harbin decreased by approximately 400,000 (Figure 1). The population of Dalian is in a state of fluctuation. In the past 5 years, the population has increased by approximately 10,000, the population of only Shenyang has slightly increased, and the urban population has increased by approximately 200,000. During the same period, the population of Hangzhou and Shanghai increased by approximately 600,000 and 250,000, respectively. The population loss phenomenon in other small- and medium-sized cities in Northeast China is considerably serious, and the entire northeast is near a state of overall shrinking. In recent years, the phenomenon of attracting young talent has appeared. Some cities provide preferential treatments to attract young people, reflecting the population problem of urban growth. In this context,

cities in Central and Western China are facing massive population loss, and numerous small and medium-sized cities are shrinking [22].

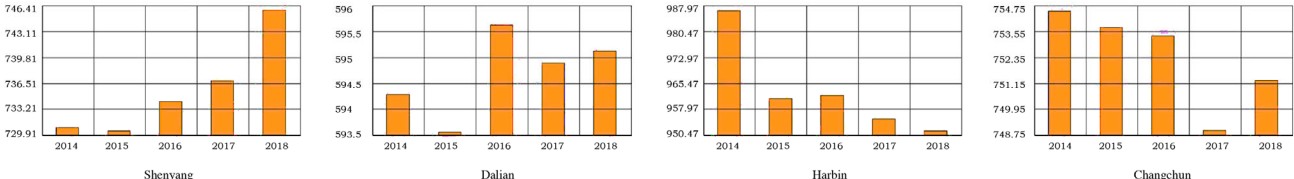

**Figure 1.** Population changes in the four major cities in Northeast China from 2014 to 2018. Source: National Bureau of Statistics.

The characteristics and demands of the construction of the urban fringe based on urban shrinking are as follows: (1) maximisation of overall benefits, including economy, society, culture, and ecology; (2) pessimism and conservatism; (3) balance of forces among all social participants; (4) abandonment of GDP doctrine; and (5) bottom-line and system thinking.

### 3.2. Influencing Factors and Interactions of Urban Expansion and Shrinking

The main factors affecting urban expansion and shrinking include population, social and economic activities, and the physical space environment. The increase and decrease in population, strength of social economic activities, and quality of the physical space environment and their interactions are used to determine the overall appearance of the urban form and movement state of shrinking inhibition or expansion [23]. Assume that the aforementioned three factors exhibit two simple states: the increase or decrease in population, the increase or decrease in social and economic activities, and the high or low quality of the physical space environment. In theory, the urban form constituting these three independent or interactive factors shows eight basic patterns (Figure 2).

The relationship among population, social and economic activities, and spatial environmental quality is complex. Some obvious direct effects of these factors are observed; for example, a good living environment and job opportunities attract many people to live and work in a city, thereby promoting population growth. When the urban space environment is poor, the capable people tend to leave the city. When the city's economy shows a decline, people consider moving to cities with development opportunities. Other situations can also occur. For example, when the economic activity increases, if the subject of the economic activity is a polluting industry, it reduces the environmental quality of the city. If economic activity has a small impact on the environment and the government can invest in environmental construction based on increased taxes, then economic activities play a positive role in improving the environment. Due to the large scale of some cities, various areas of the same city show different growth and shrinking patterns, which causes the city to present a complex coexistence of expansion and shrinking. Population elements include population quantity, population structure, and population quality. Many parameters affect the change in population and population quality, such as education, ageing, number of children, profit seeking/developmental migration, and economic development.

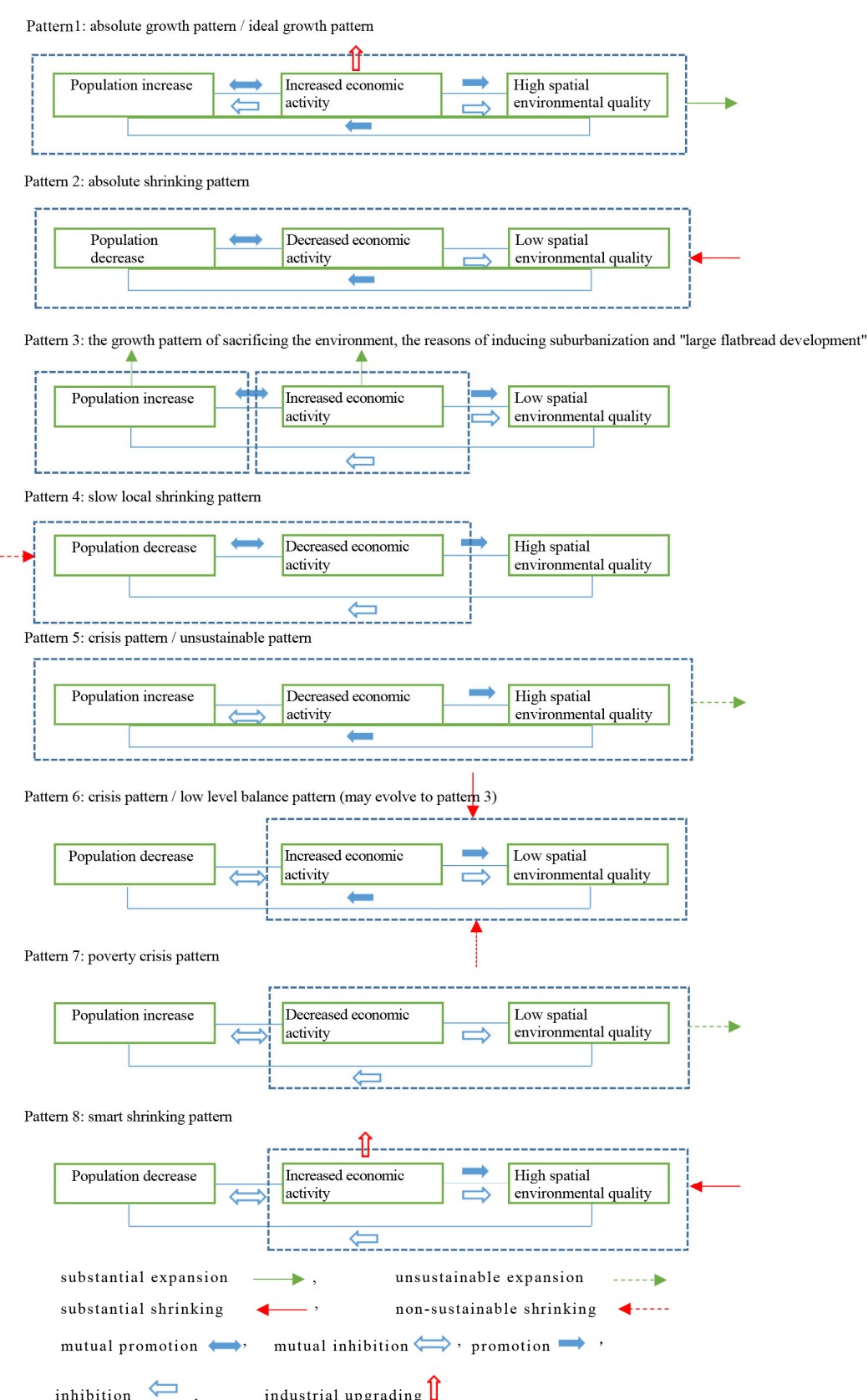

**Figure 2.** Urban growth and shrinking patterns based on the three main factors. Source: the author.

Social and economic activities include economic, social, and cultural activities. Many factors affect these activities, such as the development of science and technology, government and social investment, road traffic, nature of land property rights, consumption, resource discovery/exhaustion, industrial development/decline, population change, and global industrial and structural adjustment.

The quality of the spatial environmental includes location and ecology, that is, the location (which directly affects land prices and land property rights), geology and landforms, ecological landscape, and ecological environment. The main factors affecting spatial environmental quality include natural conditions, government investment, natural disasters, productive pollution, social damage (wars/unexpected disasters), and cultural atmosphere. If these specific factors are considered in the aforementioned patterns, each specific pattern can contain many sub-patterns (Figure 3).

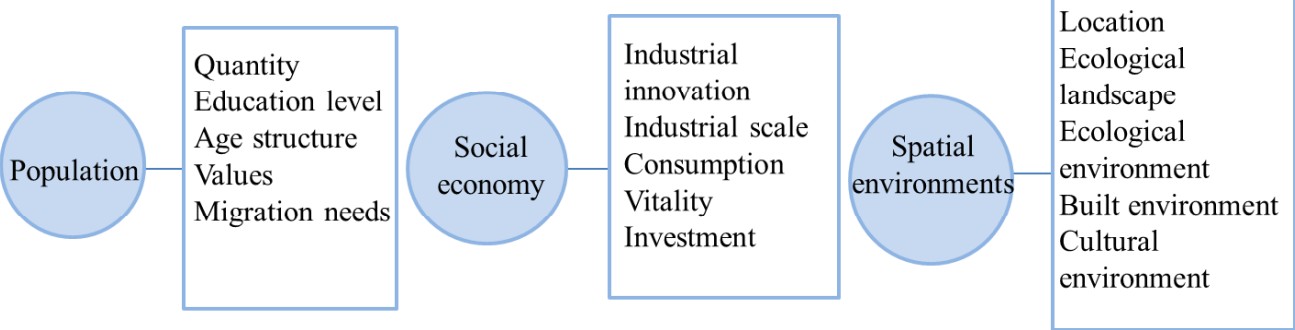

**Figure 3.** Main factors. Source: the author.

### 3.3. Urban Fringe Driven by 'Growth-Shrinking'

### 3.3.1. Urban Growth and Fringe

The aforementioned growth pattern 3 is a common pattern observed in the early stage of urban growth, and countermeasures are available for the problems caused by urban growth in both time and space. For time, countermeasures include to strengthen education and improve people's quality of life, to achieve a high-level balance among the people, economy, and environment through industrial transformation and upgrading, and to upgrade spatial environment quality. For space, considerable work must be done in terms of population, industry, and environment to solve the problems resulting from growth. Urban expansion in China and the United States is completely different for space. Facing the harsh urban space environment, the strategy of Chinese cities is to shift from a labour-intensive industry to service economy and to move numerous secondary industries to the urban fringe for ensuring the quality of urban core-area environment. In urban expansion in America, the middle class migrates to the urban fringe, escapes from the bad urban environment, and forms suburbanisation. The urban sprawl caused by suburbanisation is unsustainable, and the pattern of industries moving out has resulted in the 'large flatbread development' pattern of urban expansion, which is also unsustainable. To prevent the low-level and unrestricted expansion of cities, China's urban management starts from external factors, protects the red line of the cultivated land, strengthens ecological protection, and forcibly restricts urban growth through administrative means. Therefore, although China's urbanisation is rapid, it is not out of control. However, many problems remain in land use, ecological protection, and farmland protection, and forced restraints sometimes play a negative role in urban development. On 12 March 2020, the State Council issued 'Decision on Authorizing and Entrusting Land Use Examination and Approval Rights', which authorises the examination and approval items of transfer of the agricultural land from the permanent basic farmland to construction land authorised by the State Council to the provincial government for

approval. In this manner, local governments have considerable flexibility in dealing with urban growth. Only by comprehensively considering the population, industry, and environment and by adopting time-oriented countermeasures can the city be stabilised in an ideal state. In this manner, the urban growth pattern evolves into pattern 1, the most ideal urban growth pattern. Growth patterns 5 and 7 present unsustainable growth types due to their shortcomings in the economy, industry, and spatial environment. (Figure 4)

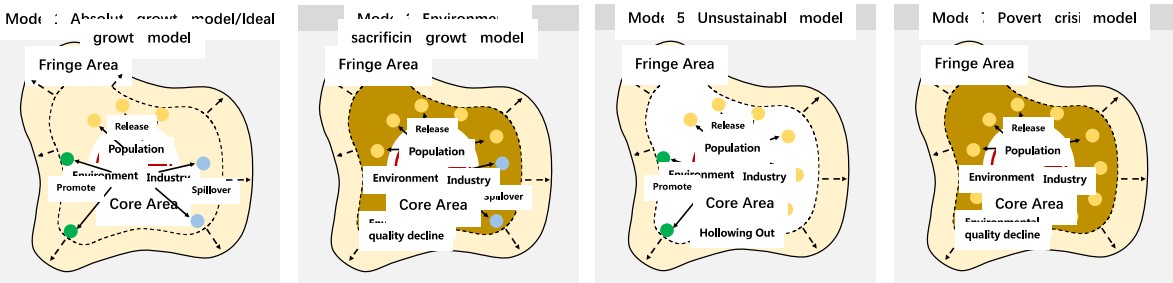

**Figure 4.** Schematic diagram of fringe area growth under four models. Source: the author.

### 3.3.2. Urban Shrinking and Fringe

Among the four urban shrinking patterns, the absolute shrinking of pattern 2 presents an almost irreversible decline without any external factors. Pattern 4 occurs in post-industrial cities, where a good urban space environment impedes city shrinking and may even stabilise the declining population at a certain level to prevent the absolute decline of the city. However, due to a continuous downturn of economic activities, the future of the city is not optimistic. The shrinking type represented through pattern 6 is the shrinking dilemma confronted by third-tier and fourth-tier cities in Central and Western China and Northeast China, where the urban space environment quality is not high, indicating that the city has not developed to a certain level. At this time, the decrease in population can have an important impact on economic development, but the governments of these cities have ignored urban shrinking and have continued to develop and expand the urban fringe area under the influence of the growth doctrine or for impeding shrinking. Some scholars have termed this situation as the 'shrinking paradox'. However, for the relationship embodied in the pattern, promoting economic development is the only approach to impede shrinking. Thus, there is no obvious problem with the selection of the government. If the investment industry achieves the reverse shrinking expansion of the urban fringe, shrinking may be restrained with industrial development promotion. However, if the reverse shrinking expansion of the urban fringe is a result of the government's promotion of real estate, then this expansion can certainly have no future. For shrinking pattern 8, economic development and good spatial environment quality can effectively prevent the shrinking trend, and this shrinking is not bad for the city. When the population decreases to a reasonable level, the city can attain a relatively high-level balance. In the growth and shrinking pattern, the economic activities and environmental quality affecting urban growth and shrinking are closely related to urban fringe areas. The location economy and ecological environment of the urban fringe are considerably active regulatory factors, which can be regulated to influence the process and effect of urban growth and shrinking. (Figure 5)

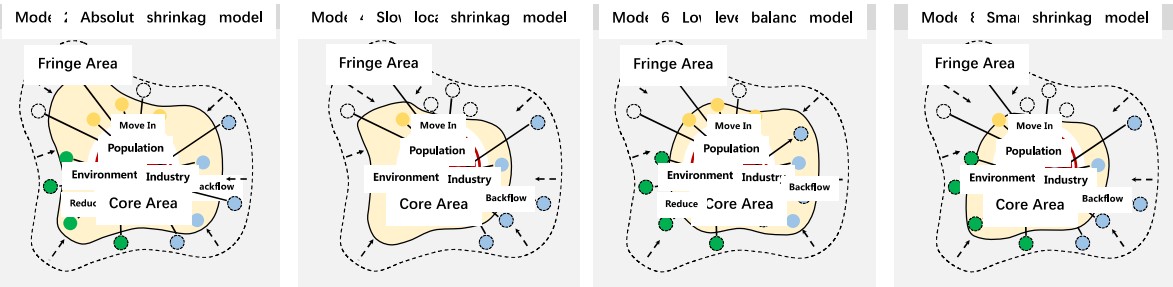

**Figure 5.** Schematic diagram of fringe area shrinkage under four models. Source: the author.

*3.4. Problems and Countermeasures*

3.4.1. Problems Faced by the Urban Fringe under the Driving Force of Urban Growth and Expansion

(1)  Severe siphon effect and severely insufficient urban functions and supply

Due to the outward expansion of the central city, the flow of people and logistics in the urban fringe continue to return to the central area, and the high-value-added industries and high-end urban functions continue to transfer to the central area. The urban fringe area does not have sufficient development motivation. This area can only depend on the advantage of land costs to develop the spillover industries in the central city area, and it cannot depend on the central city area for functional growth and upgrade [24]. For example, in the early planning of Shanghai, the development concept of 'one city and nine towns' was put forward. However, due to the siphon effect of the central city, Songjiang, Pujiang, Gaoqiao and other new towns have not been able to grow rapidly. In the initial stage, they mainly rely on residential functions, relying on low-cost land resources in exchange for human resources. As a new town in the marginal area, they cannot effectively obtain new urban functions. (Figure 6)

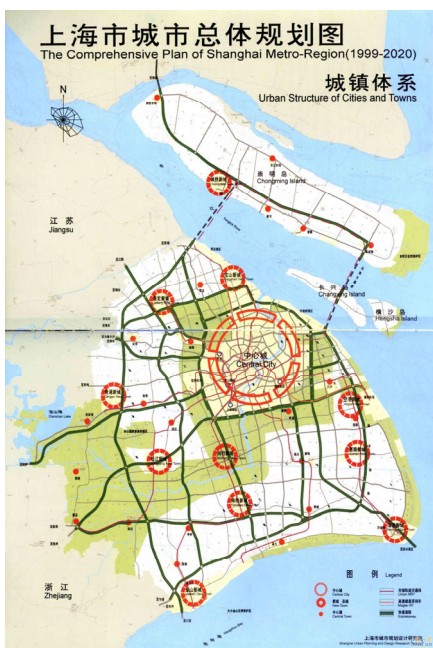

**Figure 6.** Urban system planning map of Shanghai master plan. Source: Shanghai master plan.

(2)  Difficulties in space quality improvement and urban village renovation

As the city continues to spread and expand, the rural areas and cultivated land in the urban fringe are expropriated as the urban construction land. Some villagers continue to

live in the original villages when they lose their cultivated land, thus forming a 'village in the city', which retains rural characteristics in landscape characteristics, and an urban village forms. During the expansion of central urban areas, restricted by the extensive urban management and control in the early stage, the buildings in the urban villages are constructed at will, and the external population is large; hence, overall transformation is difficult. However, the quality of the original villagers in the urban village is low, and they cannot integrate into the new urban functions of the central city, which makes living in the city in the future, relocating, and renovating difficult for the villagers. Moreover, with development, the land value continuously increases, thereby annually increasing the transformation cost. The development and construction of China's national economic zones in the early stage generally selects marginal areas. Through many years of industrialisation, urbanisation, and urban village reconstruction, the quality of cities has been comprehensively improved. However, there are certain problems, such as the slowdown of urbanisation, insufficient motivation for the renewal and transformation of urban centres in fringe areas, and low demands.

(3)   Outstanding idle land situation caused by the relocation of old factories

In the early expansion of the central city, many relocated factories and local rural enterprises have accumulated in the urban fringe. However, with expansion, the environmental costs of the urban fringe are continuously increasing and numerous industries in the area are facing secondary relocation [25], thereby making the original factory idle. The original factory must adapt to development in the background of expansion by replacing the land use function. However, due to the scattered layout of factories in the urban fringe, crisscross with urban villages, and complex ownership, transformation is difficult and the space efficiency is not met. For the fringe areas of the second and third-tier cities in China, land development has not met the required land input–output indicators. Moreover, there are generally problems, including small investment scale, low land use efficiency, and substandard input–output. Moreover, factories in some industrial parks are loosely distributed, and a large amount of land has been left idle for a long time. National industrial parks, as the highest-level industrial zones, represent the efficiency level of industrial land in China. According to the Notice on Evaluation of Land Intensive Utilization in National Development Zones in 2018 issued by the Ministry of Natural Resources in 2019, the average land tax on industrial lands in China is 6,869,500 yuan/ha. Among them, the average tax on industrial lands in the east, central region, west, and northeast is 8,334,300, 4,656,800, 5,066,800, and 4,845,700 yuan/ha, respectively. In contrast, there is still a large gap between foreign first-class industrial parks, such as Silicon Valley, and the Tsukuba Science City. (Figure 7.)

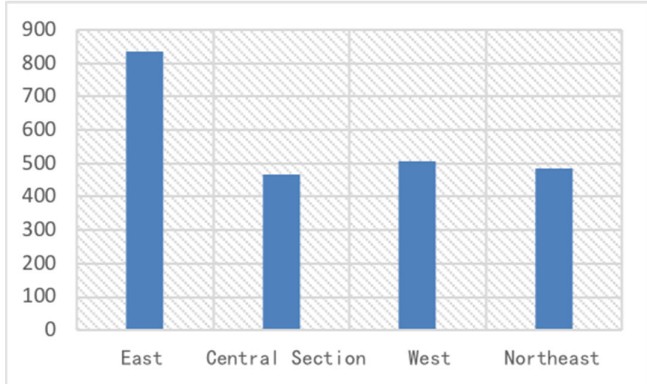

**Figure 7.** Average tax revenue of industrial land in China's national development zones in 2018. Data Source: Big database of China Business Industry Research Institute.

(4) In urbanisation, villages and towns in urban fringe areas are enthusiastic about spontaneous urbanisation. The problem of small property rights based on the collective property rights of the land has emerged, which has caused considerable problems for management and land utilisation.

3.4.2. Problems Faced by the Urban Fringe under the Driving Force of Urban Shrinking

(1) Increased costs of sharpened functions and regional integration

In the shrinking process, the control and influence of the central urban area on the urban fringe decreases. This phenomenon causes the fringe area function to focus on its development and changes in the development of fringe areas from the traditional passive to active. However, due to the influence of fluctuations in the central urban area in the early stage, the spatial development of the urban fringe lacks overall consideration and planning and is considerably uncertain. With the shrinking of the central urban area, the land renewal in the urban fringe is passively stagnated, and its spatial development is disordered [26]. Considering the Kangba New City of Ordos, Inner Mongolia, as an example of a rising new city in the fringe area, the transformation of the urban functions was highly expected[27]. Then, due to the deceleration of central urban area's development in the later period, the popularity of the Kangba New City was seriously insufficient, which could not support its own development. Eventually, it became known as a 'ghost city'.

(2) Low Intensity of Land Use

The agricultural zone in the early urban fringe was first affected through urban expansion, and its industrial structure changed from primary to secondary and tertiary industries. With the shrinking of central urban areas, the urban fringe lacked the initiative and power of land development, which caused the original industrial land, residential land, agricultural land, and other types of lands to intertwine and affect each other, forming a lose-lose situation. The industrial development was restricted, the residential environment was polluted, and the green land constantly eroded [28,29].

## 4. Strategies for the Urban Fringe

### 4.1. Update Common Demands

According to the aforementioned problems of the urban fringe, traditional market-driven development methods are difficult to adapt to the unique urban space areas of the urban fringe under the influence of spatial reconstruction. In the background of the expansion and shrinking of the spatial structure of the central urban area, to find a suitable approach to the renewal and development of the area in the new era, the changes in urban spatial form, social management pattern, and policy system and the development demands of the urban fringe should be analysed [30]. The new countermeasures in function, space, and policy should be clarified.

4.1.1. Function Demand

The urban fringe, as the connecting link between the new and old urban areas, maintains close contact and shares a part of the functions of the old urban area as the central area of the city, forming a regional specialised sub-centre dominated by production or life-supporting. In response to its own regional micro-level functional demands, the urban fringe should optimise and upgrade its functions for adapting to transformation from production- to life-oriented functions and ensure that relevant supporting and service facilities can meet the requirements of residents when the residential function constantly improves[29]. To balance the regional job vacancies resulting from the relocation of industries in the urban fringe, the original industrial functions should be screened, and some characteristic industries suitable for the development of the area should be retained.

### 4.1.2. Space Demand

During the changes in the spatial structure of the central urban area, the urban fringe should actively implement countermeasures from the aspects of the land use structure and spatial form. The first aspect is to construct the spatial structure and land use control to adapt to its functions, to increase the spatial supply of industries and characteristic functions, and to eliminate the spatial dependence on the central urban area. The second aspect is to focus on the improvement of the urban road network systems and follow up construction of supporting infrastructure, and to change the situation that the land development in the area is mainly concentrated on the main roads in the city during industrialisation, which results in the low land use efficiency within the area. The urban fringe area should optimise the land use structure through land renewal and the construction of supporting facilities, promote area transformation from the urban fringe to urban internal space, and strengthen the connection with the old urban area. The third aspect is to arrange the corresponding functions around public transportation networks that connect the new and old urban areas and to promote the transformation of the area to public transportation corridor-oriented space development. The fourth aspect is to renovate and protect the existing green space in the area by using the concept of sustainable development to make the area become a green corridor between the old and new urban areas, and to promote the overall ecological environment of the city with multi-centres.

### 4.1.3. Ecological Demand

In the urban expansion stage, different degrees of ecological damage occur and various countermeasures should be implemented. In the urban shrinking stage, these measures should be planned for implementation.

### *4.2. Renewal Differentiated Demands*

### 4.2.1. Renewal of the Urban Fringe under Urban Expansion

With the expansion of central urban areas, the urban fringe is squeezed for development power and bearing functions. Therefore, similar to Silicon Valley in the United States and Tsukuba New Town in Japan[31], during renewal, the characteristic development pattern should be highlighted in urban fringe areas, and the focus should not be on carrying the spillover function of traditional cities. The urban fringe often exhibits the characteristics of a good ecological environment and suitable development intensity. Under the expansion of cities, the urban fringe should combine its advantages to develop characteristic space and to realise the integrated development of regional networks. Through characteristic development, the urban fringe not only realises efficient development but also reduces the functional division with the central urban area and transforms from the traditional vertical and fringe to horizontal and characteristic development, thereby preventing the siphon benefit of the central urban area to the fringe area.

### 4.2.2. Renewal of the Urban Fringe under Urban Shrinking

Under the shrinking of the central urban area, the urban fringe faces problems such as population decline and insufficient vitality. Therefore, in the renewal process, the urban fringe should pay attention to the cultivation of its own power, completely depend on the characteristics of rich spatial resources and the strong environmental carrying capacity, actively attract industries with strong driving ability to settle in, and enhance the haematopoietic function. The urban fringe should further improve living-supporting facilities such as housing and catering to improve living environment quality and prevent excessive population loss.

*4.3. Countermeasures for the Urban Fringe*

4.3.1. Accurate Identification of the Spatial Scope of the Urban Fringe

In the new era, in the background of urbanisation development based on the construction of ecological civilisation, the urban fringe can no longer be characterised through scale and disorderly expansion but should transform to efficient and high-quality development. Due to the location characteristics of the urban–rural fringe and low land use efficiency, the spatial boundary of the urban fringe is fuzzy, which cannot support the stock space and accurate renewal in the new era. Therefore, combined with the development stage of the urban fringe and relationship between the surrounding urban and rural areas, the land development intensity, population concentration density, and transportation convenience should be identified to delimit the development boundary of the urban fringe and to support the spatial demand of the renewal and transformation of urban fringes.

4.3.2. Focus on the Reuse of Stock Land

In the early days, the urban fringe primarily depended on land finance and policy support, and the spatial layout was often subjected to development requirements, resulting in non-intensive land use [32]. Simultaneously, the labour-intensive and low-technology industrial system of the urban fringe must be transformed and urgently upgraded to continuously release the inefficient space. Therefore, the development and construction of stock land can become a new direction for the second development in the urban fringe. For stock land development, the urban fringe should not focus on low land costs but on the improvement of land quality and services. The urban fringe should improve the supporting service facilities around the stock land, accurately introduce the enterprises closely related to the surrounding industries and innovate the traditional development pattern of the stock land to efficiently utilise stock land.

4.3.3. Innovative Flexible Land Use Pattern

In the renewal process, the urban fringe faces the challenges of unclear direction of industrial transformation and substantial changes in enterprise types. The urban fringe should actively innovate land supply systems and land development patterns. On one hand, the urban fringe should learn from Singapore's concept of unplanted land, combine the new requirements of land space planning, and should adopt the control of unplanted land for areas with an unclear development direction to enhance the flexible supply of land through different control means such as index, space, and time blanking[31]. In specific land development, the urban fringe should learn from the development concept of M0 land use employed in Shenzhen and Guangzhou[33]; should strengthen the mixed development pattern of land use; should reduce the access threshold of innovation, R&D, and other industries; and should guide the transformation of the development pattern of the urban fringe from the land space perspective.

4.3.4. Focus on Coordination and Coexistence with New and Old Urban Areas

With the enhancement of the core position of central urban areas and the development of the urban fringe as the dominant function of the city, the functional orientation of the fringe of the connecting zone between the two should emphasise coordinated development. The urban fringe should pay attention to the connection between new and old urban areas, strengthen regional functions, and realise the coordinated development between its functions and the functions of the new and old urban areas. Simultaneously, the urban fringe should pay attention to the unified coordination of the overall area space, strengthen spatial interactions with the new and old urban areas at the macro level, focus on the organisational structure of the internal space and protection of the entire ecological space of the area at the micro level, and ensure

the spatial integrity and sustainable development of the area through reasonable active guidance of land renewal.

### 4.3.5. Focus on the Improvement of Supporting Facilities

Because supporting facilities in the urban fringe passively undertake the functions of the central urban area, the urban fringe should first improve the road network structure of the area to develop a multi-level and networked road system. The urban fringe should strengthen the construction of internal public construction supporting facilities, especially of the 15-minute living circle, strengthen basic livelihood engineering facilities, and form high-quality public spaces to ensure that the service radius of supporting facilities can cover the entire area and satisfy the daily production and living requirements of residents in the area.

### 4.3.6. Focus on the Planned Transformation of Urban Villages

For the problems faced during the land renewal of urban villages in the urban fringe, the adoption of simple and crude methods inevitably leads to many social problems. The urban fringe should formulate a reasonable transformation plan to successively change the urban villages in the area in batches. Simultaneously, on the basis of maintaining the role of accommodating the foreign population, the urban fringe should implement the resettlement of residents of original urban villages to reduce the impact of urban village transformation on urban labour supply to the maximum extent [34].

### 4.3.7. Focus on the Sustainable Land Renewal Development

For the aforementioned problems, such as idle land, low intensive land use, and serious damage to the ecological environment caused by the relocation of old industrial areas in the urban fringe during spatial structure transformation in central urban areas, the urban fringe should adhere to bottom-line thinking; should not be expanded blindly [35]; should involve the reuse of abandoned industrial, residential, and agricultural land; and in urban renewal, should involve the complete use of the original resources, reconstruction of industrial and residential buildings, improvement in the utilisation rate of all types of urban construction land, saving of investment, and reducing the generation of construction waste.

## 5. Conclusions

The study focused on the construction of spatial environmental quality in the urban fringe, the leading role played by location that provides complete advantage, the development of the ecological environment and ecological landscape, and industrial upgrading promotion. The study solved the specific problems caused by growth and shrinking, achieved the balance between the acquisition and provision of social and natural resources in the urban fringe, and promoted the healthy and sustainable development of cities in the dynamic growth and shrinking changes.

Based on the 'expansion–contraction' dynamics model, this study investigated the problems faced by fringe areas during contraction and growth. These problems included the interaction between the three major factors of residence, industry, and environment in the model and the effects and consequences of different modes. In addition, countermeasures that can be adopted for the future development of the fringe area based on the aforementioned study are proposed. As the most sensitive area for urban space expansion and contraction, the fringe area has the advantages of low land cost and superior environmental base and the shortcomings of mixed urban and rural functions. In development, confronting problems such as insufficient power and obvious siphon effect is easy. In view of this, three aspects of high popularity, superior industry, and excellent environment should be emphasised. Attention must be paid to the construction of a high-quality space environment and to the advantages of location and the leading

role of the ecological environment and landscape for promoting industrial upgrading and solve specific problems resulting from growth and contraction. Ultimately, the fringe area can achieve a balance between social and natural resources, which is conducive to the healthy and stable development of cities in the dynamic growth and contraction changes.

This study constructed a dynamic model of growth and contraction based on three main parameters of population, space environment, and industry. Eight growth–contraction framework models were obtained to analyse and classify the evolution of urban fringe areas. China's urban fringe areas are in a period of coexistence of growth, contraction, and stagnation (stability). In addition, the urban development strategy proposed by the Chinese government has gradually changed from single to multiple. For the development of China's fringe areas, the most important thing is to clarify the contraction–growth model and to formulate corresponding strategies based on model characteristics to avoid further problems.

This study only constructed a framework model (Figure 8). Subsequent in-depth research must refine the model to obtain the complexity and diversity of urban growth and contraction and further formulate corresponding strategies on the development of urban fringe areas. For population parameters, the framework model only includes urban development according to the changes in population. However, the population changes in terms of not only quantity but also quality, such as education level, which is not reported in this paper. Similarly, the types of economy are diverse, and the government's control over land, space environment, and urban boundaries are various and complex. Therefore, an empirical study must be conducted using relatively more in-depth

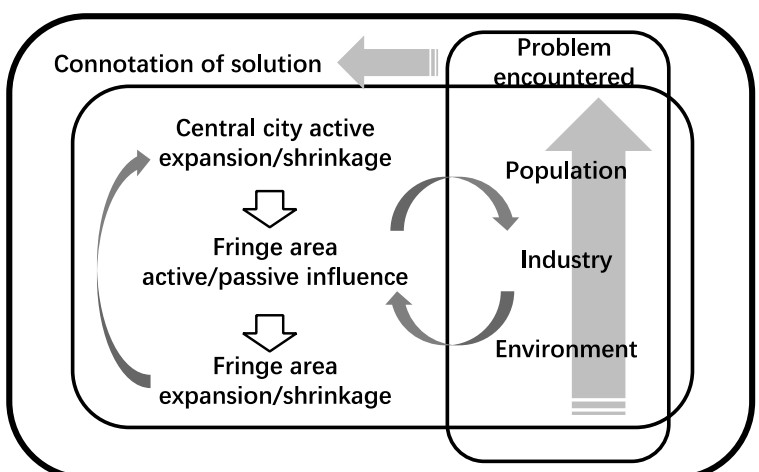

**Author Contributions:** Conceptualization, C.S. and W.L.; methodology, C.S.; software, C.S.; validation, C.S.; formal analysis, C.S. and W.L.; investigation, C.S.; resources, C.S. and W.L.; data curation, C.S.; writing—original draft preparation, C.S.; writing—review and editing, C.S. and W.L.; visualization, C.S..; supervision, C.S.; project administration, W.L.; funding acquisition, C.S. All authors have read and agreed to the published version of the manuscript.

**Funding:** This research received no external funding.

**Institutional Review Board Statement:** Not applicable, this research did not involve humans or animals.

**Informed Consent Statement:** Not applicable, this research did not involve humans or animals.

**Data Availability Statement:** The original contributions presented in the study are included in the article/supplementary materials. Further inquiries can be directed to the corresponding author/s.

**Conflicts of Interest:** The authors declare no conflict of interest.

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
