# Peer review of "Study on the Urban Fringe Based on the Expansion–Shrinking Dynamic Pattern"

_sustainability, doi:10.3390/su13105718_

Round 1

Reviewer 1 Report

  1. There are several arguments that are not supported by literature
  2. No clear methodology visible
  3.  Empirical research not visible
  4.  Absence of graphics that could turn more evident the argument
  5. Conclusions are almost inexistent (6 lines).

Author Response

Dear Reviewer,

Thank you very much for your valuable guidance. I have revised the paper according to your opinions. The specific contents are as follows:

Point 1: There are several arguments that are not supported by literature.

Response 1: In order to solve this point, the paper strengthened the citation of existing research results in the new content, and added 19 points of view of the paper. See the text 45\79\81\159\162\163\165\168\188\189\196\197\210\222\223\329\368\369\370\608-614\624-626\631-656\671-685 for details.

Point 2: No clear methodology visible

Response 2: For this point, the paper adds the main structure diagram of the paper, the logical structure and methods of the paper, and systematically describes the main idea of the paper. See Figure 8. Text 560-584\709-711 for details;

Point 3: Empirical research not visible

Response 3: In view of this point, the paper introduces specific examples in the problem description, and increases empirical research based on the development experience of domestic cities. See  330-335\348-353\362-374\389-394 for details;

Point 4: Absence of graphics that could turn more evident the argument

Response 4: In order to solve this point, the paper adds five pictures to describe the contraction of the edge area, the expansion of the edge area, and the land efficiency of the economic zone. See Figure 4-7 for details;

Point 5: Conclusions are almost inexistent (6 lines).

Response 5: To solve this problem, the conclusion has been rewritten, See line 560-594 for details.

The above is all the modification content, thank you very much for your patient guidance!

Best wishes!

Sui Changqing

School of Architecture &Fine Art

Dalian University Of Technology

E-mail: scq@maiI.dlut.edu.cn

Reviewer 2 Report

  1. The study is interesting and significant but this comprehensive is lacking the figures or images for more understanding and references.
  2. There is lack of citation and method of study to prove the concepts in paper logically and scientifically.
  3. There is minor improvement required in English spellings.

Author Response

Dear Reviewer,

Thank you very much for your valuable guidance. I have revised the paper according to your opinions. The specific contents are as follows:

Point 1: The study is interesting and significant but this comprehensive is lacking the figures or images for more understanding and references.

Response 1: In order to solve this point, the paper adds five pictures to describe the contraction of the edge area, the expansion of the edge area, and the land efficiency of the economic zone. See Figure 4-7 for details

Point 2: There is lack of citation and method of study to prove the concepts in paper logically and scientifically.

Response 2: Adds the main structure diagram of the paper, the logical structure and methods of the paper, and systematically describes the main idea of the paper. See Figure 8. Text 560-584\709-711 for details; and the paper strengthened the citation of existing research results in the new content, and added 19 points of view of the paper. See the text 45\79\81\159\162\163\165\168\188\189\196\197\210\222\223\329\368\369\370\608-614\624-626\631-656\671-685 for details.

Point 3: There is minor improvement required in English spellings.

Response 3: For this point, MDPI's English editors have improved the language of the article.

The above is all the modification content, thank you very much for your patient guidance!

Best wishes!

Sui Changqing

School of Architecture &Fine Art

Dalian University Of Technology

E-mail: scq@maiI.dlut.edu.cn

Round 2

Reviewer 1 Report

I consider that the paper is publishable.